

# Measurement of NO and NH₃ Concentrations in Atmospheric Simulation Chamber Using Direct Absorption Spectroscopy

Nakwon Jeong[1,2], Seungryong Lee[1,2], Soonho Song[2], Daehae Kim[1], Miyeon Yoo[1], Changyeop Lee[1]

[1]Korea Institute of Industrial Technology (KITECH), 89 Yangdaegiro-gil, Cheonan, 31056, Republic of Korea

[2]Department of Mechanical Engineering, Yonsei University, 50 Yonsei-ro, Seoul, 03722, Republic of Korea

*Correspondence to*: Miyeon Yoo (myyoo@kitech.re.kr), Changyeop Lee (cylee@kitech.re.kr)

**Abstract.** In urban atmospheric chemistry, nitrogen oxides and ammonia in the atmosphere are major species participating in the secondary aerosol formation process, causing severe environmental problems such as decreased visibility and acid rain. In order to respond effectively to particulate matter problems, the correlation of precursors should be identified in detail. This

study used UV-C light to convert gaseous substances into particulate substances in the atmospheric simulation chamber to simulate the photochemical reaction. The effects of several operating variables, such as UV-C light intensity, relative humidity, and initial concentrations of $O_2$, NO, and $NH_3$, on the $NH_4NO_3$ formation were investigated. Since atmospheric gas species are short-lived, they require a measurement technique with an ultra-fast response and high sensitivity. Therefore, the concentrations of NO and $NH_3$ were measured using Direct Absorption Spectroscopy techniques with the wavenumber regions

of 1926 and 6568 cm$^{-1}$, respectively. NO and $NH_3$ were precisely measured with an error rate of less than 3% with the reference gas. The results show that NO and $NH_3$ were converted over 98% when UV-C light intensity was 24W and relative humidity was about 30% at 1 atm, 296 K. It also showed that higher UV-C light intensity, $O_3$ concentration, and relative humidity induced higher conversion rates and secondary aerosol generation. In particular, it was experimentally confirmed that the secondary aerosol generation and growth process was greatly influenced by relative humidity.

**1 Introduction**

Aerosols in the atmosphere cause severe environmental and social issues by adversely affecting human health, disrupting solar radiation transfer, and influencing cloud formation (Zhang et al., 2015; Butt et al., 2016). Airborne aerosols are divided into two groups. Primary aerosols are emitted directly from various emission sources, and secondary aerosols are formed via homogeneous nucleation through physicochemical processes in the atmosphere. Precursor materials such as nitrate, sulfate,

and ammonia in the atmosphere are major species that participate in secondary aerosol formation and growth processes. Nitrogen oxides contribute to acid deposition and form atmospheric oxidants ozone and nitrate radical (Fuchs et al., 2010). Ammonia increases natural water and soil acidity, providing excessive nitrogen injection into the ecosystem (Stevens C. J. et al., 2010). Sulfate or nitrate, which accounts for a large proportion of the contribution to airborne aerosol production, leads to localized cooling by scattering solar radiation (Adopted, 2014). It can also influence the macroscopic and microscopic physical



properties of clouds and affect precipitation variability by providing a source of cloud condensation nuclei (CCN) or ice nuclei (IN) (Tao et al., 2012).

These aerosols are the leading causes of respiratory diseases and exacerbation, and the prevalence of chronic obstructive pulmonary disease (COPD) is reported to increase by 33% when the concentration of aerosol increases by 7 μg/m$^3$ in five years (Schikowski et al., 2005). In addition, short-term exposure to aerosols increases lung inflammation, worsening

respiratory symptoms, and chronic respiratory diseases such as COPD and lung cancer occur when exposed in the long-term (World Health Organization, 2017). Since the impact on human health varies considerably depending on aerosol size, concentration, and composition, research on the interaction of precursors during the production process is required.

Many researchers measured and analyzed pollutants in the atmosphere to study the interaction of precursors in detail and modeled and simulated based on them. The conversion rate of $SO_2$ and $NO_2$ according to $O_3$ concentration and relative

humidity was measured in summer and winter (Khoder, 2002), and the HOx behavior in urban areas in winter was investigated by measuring OH and $HO_2$ concentrations, and the atmospheric oxidation process was studied (Xinrong et al., 2006). In addition, studies were conducted to investigate the relationship between $O_3$, NO, and $NO_2$ concentration in Tianjin (Han et al., 2011) and the role of $NH_3$ in the ionic chemistry of $PM_{2.5}$ in wintertime (Zhao et al., 2016). Air pollution models have been developed for decades to predict how pollutants behave in the atmosphere and to describe the relationships between emissions,

gaseous substances, and particulate matter. Harkonen et al. (1996) developed a CALINE4 model to predict air pollution concentrations near roads, and Kelly et al. (2018) evaluated the Community Multiscale Air Quality (CMAQ) model through air quality measurement and predicted future trends of $NH_4NO_3$.

However, since many factors, such as meteorological conditions and pollutant emission rates, cannot be artificially controlled, it is difficult to determine the cause of secondary pollutants by monitoring only the concentration of reactants and products in

complex photochemical chemistry (Lee et al., 2009). Consequently, studies have been conducted actively using atmospheric simulation chambers that can study the nucleation, hygroscopic growth, and extinction of specific compounds in an environment isolated from the outside. Spicer (1983) studied in a large-scale atmospheric simulation chamber (17.3 m$^3$) to understand the atmospheric NOx conversion rates and the relationship between nitrates, hydrocarbons, and NOx precursors. Besides, studies were conducted to investigate the secondary inorganic aerosol (SIA) formation kinetics in an outdoor

simulation chamber and to investigate the effects of NOx and $NH_3$ on secondary organic aerosol (SOA) formation from photooxidation of toluene in the indoor chamber (Behera and Sharma, 2011; Qi et al., 2020).

In most atmospheric simulation chamber studies, the concentration of gaseous precursors was measured using sampling methods such as chemiluminescence, FTIR, and NDIR. These sampling methods have a long time lag due to the sampling and analyzing process, so it is difficult to trace the rapidly fluctuating photochemical reaction. Moreover, the analysis methods are

insufficient since they are contact types and can interfere with gaseous precursors' reactions. Tunable Diode Laser Absorption Spectroscopy (TDLAS) is an optical measurement method that enables quantitative concentration measurement, instantaneous response, non-intrusive type, high precision, and reliability. It is used by continuous monitoring methods to measure the concentration of specific gaseous species, temperature, pressure, and gas velocity in various environments such as combustion



and the atmosphere. Therefore, it is a suitable method for measuring the concentration of precursors in real-time during gas-

to-particle conversion processes.

To effectively cope with airborne aerosol, it is essential to understand the correlation of precursors with various variables in detail. However, there are still many uncertainties about the mechanism associated with nucleation and hygroscopic growth of aerosols and a shortage of research on the transient analysis of the photochemical aging process. For instance, it is well known that the oxidation reaction between NO and $O_3$ results in the formation of $NO_2$. However, there is still insufficient research on

how much it reacts according to surrounding environmental conditions such as UV light intensity, relative humidity, and temperature, and how much the conversion rate or rate constant is. In this study, the $NH_4NO_3$ formation process through photochemical reactions is simulated in the indoor atmospheric simulation chamber with several variables, and the concentrations of NO and $NH_3$ precursors are measured and analyzed simultaneously using the TDLAS technique. This study can improve understanding of the roles of UV light intensity, relative humidity, and concentration rates of $O_3$, NO, and $NH_3$

in the $NH_4NO_3$ formation process by photochemical reactions. Furthermore, it is possible to analyze the dominant reactions, inhibitory characteristics, and reaction acceleration causes of the observed factors under various conditions.

## 2 Theoretical background

In the realm of modern optical technology, absorption stands as a crucial phenomenon for the quantitative sensing of thermal, physical, and molecular properties. Absorption occurs when a light source of a specific frequency passes through a gas medium

and resonates with the absorption transition of the gas species. Direct absorption spectroscopy (DAS) is a powerful tool that can be employed to measure concentration, temperature, and pressure, which are essential physical factors in chemical kinetics, using the absorption generated as described above. The measurement principle of DAS can be explained by Eq. (1) based on the Beer-Lambert Law.

$$\left(\frac{I_t}{I_0}\right)_v = exp\left(-\alpha_v\right) = exp(-k_v \cdot L) \tag{1}$$

Here, $I_t$ and $I_0$ represent transmitted light intensity and incident light intensity, respectively, and fractional transmission $\tau_v$ can be expressed as the ratio thereof (Chao et al., 2012). And $\alpha_v$ is absorbance, $k_v$ $(cm^{-1})$ is spectral absorption coefficient at frequency $v$ $(cm^{-1})$, $L$ $(cm)$ is optical path length. The spectral absorption coefficient is further defined by

$$k_v = \sum_{i,j} S_{i,j}(T) \cdot \phi\left(v - v_{0,i,j}\right) \cdot P \cdot X_i \tag{2}$$

As shown in Eq. (2), for a single transition $j$ of a specific gas species $i$, the spectral absorption coefficient $k_v$ is defined by the

product of line strength of a particular absorption line $S_{i,j}(T)$ $(cm^{-2}atm^{-1})$, line-shape function $\phi(v - v_{0,i,j})$ $(cm)$, the total pressure of the gas mixture $P$ $(atm)$, mole fraction $X_i$ according to gas temperature $T$ $(K)$ (So et al., 2022).

$$A_i = \int_{-\infty}^{\infty} \alpha_v \, dv = S_{i,j}(T) \cdot P \cdot X_i \cdot L \tag{3}$$



As described in Eq. (1), absorbance $\alpha_v$ can be expressed as the product of the spectral absorption coefficient $k_v$ and the optical path length $L$. Since the line-shape function is normalized to $\int \phi(v - v_{0,i,j})dv = 1$, the integrated absorbance area $A_i$ can be expressed as Eq. (3). Consequently, if the pressure, optical path length, and line strength are determined, the mole fraction of a specific gas species $i$ can be measured using a DAS.

## 3 Experimental setup

### 3.1 Atmospheric simulation chamber

This study was investigated the nucleation and hygroscopic growth process of $NH_4NO_3$ in real-time by measuring the concentrations of NO and $NH_3$ in an environment isolated from the outside. The atmospheric simulation chamber was designed with cylindrical quartz with a volume of ~ 3.85 L and a surface-area-to-volume ratio of ~ 1.75 m$^{-1}$. Since the quartz material used in the experiment has a low UV transmittance, a total of 4 UV-C (100 to 280 nm) lamps (Hansung Ultraviolet Company GHO436T5VH) were installed inside the chamber. These lamps were crucial in inducing the photochemical reaction.

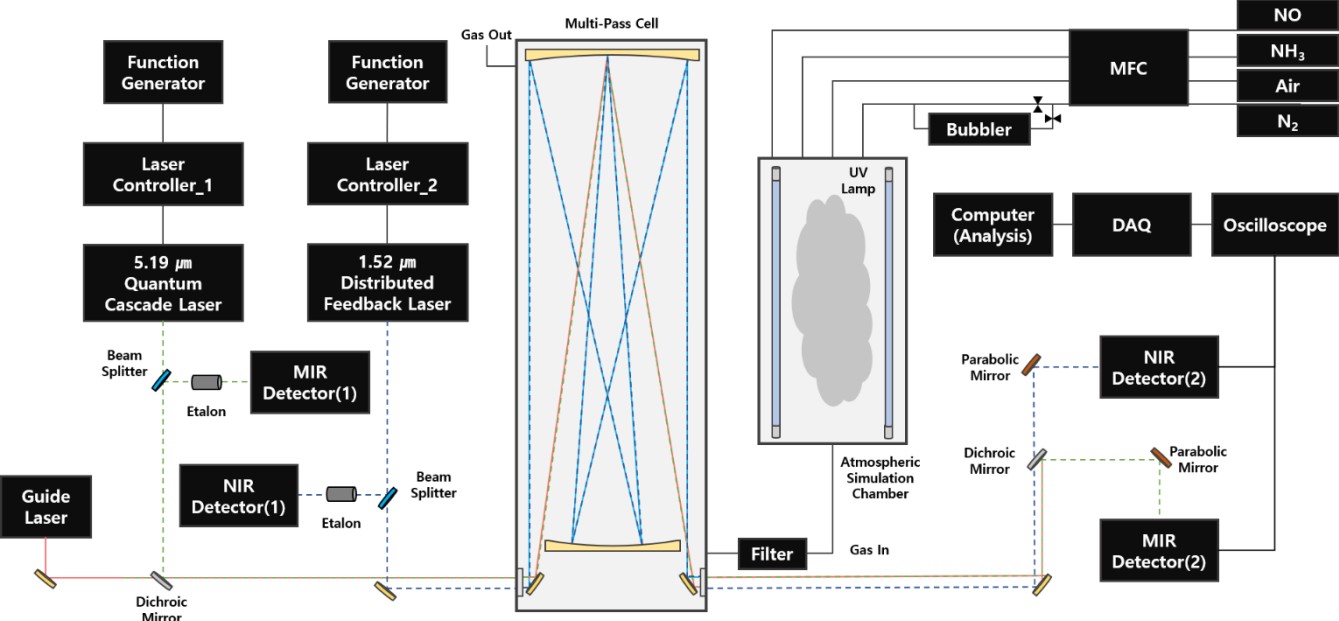

**Figure 1: Schematic diagram of atmospheric simulation system for inducing photochemical reactions and an optical sensor system for measuring gaseous precursor concentrations; MFC: mass flow controller, DAQ: data acquisition, MIR: mid-infrared, NIR: near-infrared**

Fig. 1 shows a schematic diagram of an overall experiment combining the atmospheric simulation system part to induce the photochemical reaction and the optical sensor system part to measure precursor concentration. The external environment is





maintained at 1atm, $296 \pm 2$ K, and 30 to 60% relative humidity, which are typical conditions for atmospheric simulations. The experiment used purified air, high-purity $N_2$, and 300 ppm of $NO/N_2$, $NH_3/N_2$ reference gas supplied through a calibrated mass flow controller (MFC). In order to control relative humidity in the chamber, high-purity $N_2$ was passed through the $H_2O$

bubble generator heated to about 350 K and supplied to the chamber. When $N_2$ was injected at $2\,L/min$, the relative humidity was measured at 60%. The temperature in the chamber was maintained at $296 \pm 2$ K. To minimize adsorption at the gas line, the distance between the simulation chamber and the multi-pass cell was placed close, and all reactants were continuously flowed to ensure thorough mixing within the chamber. After 30 seconds, the UV-C lamp is activated. Also, high-purity $N_2$ was flushed at $1\,L/min$ for at least 1 hour before and after the experiment to remove gas species adsorbed on the inner wall of the

chamber. The polytetrafluoroethylene (PTFE) microporous membrane filter with a pore size of 1.0 μm was used to sample $NH_4NO_3$ formed by the photochemical reaction.

### 3.2 Laser sensor system setup

The optical sensor system can be divided mainly into the laser transmitter and absorption signal receiver. The high heat load (HHL) type quantum cascade (QC) laser of 1926 cm$^{-1}$ spectral range (Alpes Lasers, Switzerland) and the butterfly 14 pin type

of distributed feedback (DFB) laser of 6568 cm$^{-1}$ spectral range (NTT Electronics, Japan) were installed as shown Fig. 1. The laser controllers (Arroyo Instruments 6310-QCL, ILX Lightwave LDC-3908) that apply a constant temperature and current was used to emit the laser with appropriate wavenumber. Here, the temperature and current of the laser controllers were verified to confirm the emitting selected spectral range using a wavemeter (HighFinesse Laser and Electronic Systems WS6-200) calibrated with a stabilized laser reference (HighFinesse Laser and Electronic Systems SLR 1532). 10 kHz ramp wave was

applied using a function generator (Tektronix AFG31000) that regulates voltage and frequency. The light source emitted from the laser was split at 9:1 by a beam splitter, and the minor light passed through the solid etalon to convert the time domain to the wavenumber domain. To compensate for the relatively weak absorption line strength of $NH_3$, a white-type multi-pass cell with a length of 0.5 m and a total path length of 25 m was used. On the contrary, in the case of NO, the absorption line strength is stronger than that of $NH_3$, so the path length was aligned to 1 m. In the absorption signal receiver part shown in Fig. 1, the

light source that passed the multi-pass cell was focused through a parabolic mirror and then irradiated to the amplified photodetectors (Thorlabs PDAVJ8, PDA50B2) to minimize optical loss. The measured absorption signal can be visualized via oscilloscope (Teledyne WS3024z), and the signals acquired by the data acquisition (DAQ) system with 10 MSs$^{-1}$ and analyzed using MATLAB.

### 3.3 Experimental conditions

The process of nucleation and hygroscopic growth of secondary aerosols by photochemical reactions in the atmosphere is greatly influenced by the atmospheric conditions. These experiments were conducted to investigate the effect of each variable because it reacts sensitively to the process of formation and growth of aerosols depending on UV light intensity, relative humidity, and concentration of secondary aerosol precursors.



**Table 1: Experimental conditions for photochemical reaction simulation.**

| | Case | 1 | 2 | 3 | 4 | 5 |
|---|---|---|---|---|---|---|
| Mass flow rate [slpm] | NO (300 ppm) | 3 | 3 | 3 | 2 / 2.5 / 3 / 3.5 / 4 | 3 |
| | NH$_3$ (300 ppm) | 3 | 3 | 3 | 3 | 2 / 2.5 / 3 / 3.5 / 4 |
| | Purified Air | 1 | 0 / 0.5 / 1 / 1.5 / 2 | 0 / 0.5 / 1 / 1.5 / 2 | 1 | 1 |
| | N$_2$ (99.99%) | 1 | 2 / 1.5 / 1 / 0.5 / 0 | 2 / 1.5 / 1 / 0.5 / 0 | 2 / 1.5 / 1 / 0.5 / 0 | 2 / 1.5 / 1 / 0.5 / 0 |
| Bubble generator | | X | X | O | X | X |
| UV-C lamp | | 1 / 2 / 3 / 4 | 2 | 2 | 2 | 2 |

The experimental conditions are summarized in Table 1. In case 1, the number of UV-C lamps was adjusted to compare the concentration reduction and conversion rate of NO and NH$_3$ according to the UV light intensity. In case 2, the mass flow rate of purified air and N$_2$ was controlled and measured to confirm the photochemical reaction of NO and NH$_3$ according to the concentration of O$_2$, and case 3 was conducted to check the effect of relative humidity in the NH$_4$NO$_3$ formation by photochemical reactions. Through cases 4 and 5, the effect of NH$_4$NO$_3$ formation depending on the mixing ratios of NO and NH$_3$ was investigated.

# 4 Results and discussion

## 4.1 Absorption simulation and line selection

In order to accurately measure the concentrations of gaseous NO and NH$_3$, which are secondary aerosol precursors, it is important to select a single absorption line without absorption interference from other gas species and sufficient line strength. Interference with other gas species can occur differently depending on ambient conditions such as temperature, pressure, and concentration. For instance, if the temperature or pressure of the ambient environment is high, the absorption linewidth generally increases, so there can be interference even if the absorption line strength is separated from other gas species at standard conditions. In addition, if other gas species have an absorption line strength around the selected scan range and their mole fraction is high, this also causes interference. Moreover, when selecting the absorption wavenumber in the mid-infrared region, caution must be taken in order to avoid interference and increased overlap with other absorption signals due to the increase in full width at half maximum (FWHM) with decreasing absorption wavenumber.

In the case of NO, the absorption line strength in the 1700 to 2000 cm$^{-1}$ spectral range is about $10^3$ times stronger than that of the near-infrared region of 5800 cm$^{-1}$, which is advantageous for precise measurements. Fig. 2(a) shows the absorption line strength of reactants and products from NH$_4$NO$_3$ photochemical chemistry in the 1700 to 2000 cm$^{-1}$ spectral range based on the HITRAN2016 database (Gordon et al., 2017). However, it is difficult to measure because of the interference of H$_2$O and NH$_3$, which are NH$_4$NO$_3$ photochemical reactants, under the 1900 cm$^{-1}$ spectral range. Fig. 2(b) shows the simulation results under experimental conditions in the 1900 to 1940 cm$^{-1}$ spectral range at standard temperature and pressure (STP). In the



spectral range from 1910 to 1925 cm⁻¹, NO absorbance is interfered with by $H_2O$, and the absorption line strength is weakened as the wavenumber increases in that region. Consequently, the 1926 cm⁻¹ spectral range with a strong line strength of about $5.16 \times 10^{-20} \ cm/mol$ while avoiding interference with other gas species was selected.

**Figure 2: (a)** Absorption line strength of reactants and products from $NH_4NO_3$ photochemical chemistry in the 1700 ~ 2000 cm⁻¹ spectral range, **(b)** simulated absorbance spectrum of $X_{H2O}$ 0.1%, $X_{NO}$ 100 ppm in the 1910 ~ 1940 cm⁻¹ spectral range, **(c)** Absorption line strength of reactants and products from $NH_4NO_3$ photochemical chemistry in the 4000 ~ 7000 cm⁻¹ spectral range, **(d)** simulated absorbance spectrum of $X_{H2O}$ 0.1%, $X_{N2O}$ 10 ppm, $X_{NH3}$ 100 ppm in the 6560 ~ 6580 cm⁻¹ spectral range at 1atm, 296 K based on HITRAN2016 database.

$NH_3$ absorption line strength is distributed over a wide range. The absorption line strength of the mid-infrared region is about 30 times stronger than the near-infrared region, and the interference with $H_2O$ decreases under the 1250 cm⁻¹ spectral range. However, due to the broad scan range of the laser in the mid-infrared region, where $NH_3$ absorbance is mainly distributed,





causes interference with $NH_4NO_3$ photochemical reactants. On the other hand, the absorption line strength is relatively weak

185 in the near-infrared region. However, it effectively avoids interference from other gases, such as $H_2O$, because it has a narrow

linewidth. Furthermore, relatively weak absorption line strength can be compensated by using a multi-pass cell to increase the

optical path length. As shown in Fig. 2(c), $NH_3$ has an absorption line strength distribution in the 4000 to 7000 $cm^{-1}$ spectral

range. As the wavenumber of a tunable diode laser decreases below the 5000 $cm^{-1}$ spectral range, the laser output power also

decreases. Accordingly, when a wavenumber region of 5000 $cm^{-1}$ or less is selected, the signal-to-noise ratio (SNR) may be

190 lowered when a multi-pass cell is used, and thus, the uncertainty may increase. In addition, there is no interference from other

gas species in the region of 6000 $cm^{-1}$, but it is not suitable for precise measurement compared to the 6500 $cm^{-1}$ spectral range

because it has an absorption line strength of less than $1.0 \times 10^{-21}\ cm/mol$. Fig. 2(d) shows the simulation results when the

concentrations of $H_2O$, $N_2O$, and $NH_3$ are 0.1%, 10 ppm, and 100 ppm, respectively, and the optical path length of 25 m in the

6560 to 6580 $cm^{-1}$ spectral range at STP. In most regions, the absorption line strengths are weak and overlap, so the 6568 $cm^{-1}$

spectral range, which is more than twice as strong as the nearby other absorption line strengths, was selected.

## 4.2 Photochemical reactions in the atmospheric simulation chamber

This study was conducted in an atmospheric simulation chamber to artificially control the ratio of secondary aerosol precursors

and atmospheric conditions such as UV light intensity and relative humidity. The energy emitted from the UV-C lamps in the

chamber, which is filled with purified air, nitric oxide, and ammonia, caused numerous chemical reactions. Table 2 shows the

primary photochemical reactions participating in the $NH_4NO_3$ formation process and the rate constant of each reaction.

**Table 2: The primary photochemical reactions and rate constant participating in the $NH_4NO_3$ formation process for Ox, HOx, and NOx.**

| Reaction number | Photochemical reactions | Rate constant, 298K $[cm^3 \cdot molecule^{-1} \cdot s^{-1}]$ | Reference |
|---|---|---|---|
| (4) | $O_2 + \cdot O + M \rightarrow O_3 + M$ | $5.6 \times 10^{-34}$ | |
| (5) | $\cdot O + O_3 \rightarrow 2O_2$ | $8.0 \times 10^{-15}$ | |
| (6) | $\cdot OH + O_3 \rightarrow \cdot HO_2 + O_2$ | $7.3 \times 10^{-14}$ | |
| (7) | $\cdot O + \cdot HO_2 \rightarrow \cdot OH + O_2$ | $5.8 \times 10^{-11}$ | |
| (8) | $\cdot OH + \cdot OH \rightarrow H_2O + \cdot O$ | $1.48 \times 10^{-12}$ | |
| (9) | $\cdot O + NO + M \rightarrow NO_2 + M$ | $3.0 \times 10^{-11}$ | |
| (10) | $\cdot O + NO_2 \rightarrow O_2 + NO$ | $1.0 \times 10^{-11}$ | |
| (11) | $\cdot O + NO_2 + M \rightarrow NO_3 + M$ | $2.3 \times 10^{-11}$ | |
| (12) | $\cdot OH + \cdot HONO \rightarrow H_2O + NO_2$ | $6.0 \times 10^{-12}$ | |
| (13) | $\cdot OH + NO + M \rightarrow \cdot HONO + M$ | $3.3 \times 10^{-11}$ | |
| (14) | $\cdot OH + NH_3 \rightarrow H_2O + NH_2$ | $1.6 \times 10^{-13}$ | |
| (15) | $NO_2 + H_2O \rightarrow \cdot HONO$ | $2.4 \times 10^{-23}$ (variable) | |
| (16) | $NO + NO_2 + H_2O \rightarrow \cdot HONO + \cdot HONO$ | $6.0 \times 10^{-38}$ | |



| | | $[cm^6 \cdot molecule^{-2} \cdot s^{-1}]$ |
|---|---|---|
| (17) | $\cdot OH + NO_3 \rightarrow \cdot HO_2 + NO_2$ | $2.0 \times 10^{-11}$ |
| (18) | $\cdot HO_2 + NO \rightarrow \cdot OH + NO_2$ | $8.8 \times 10^{-12}$ |
| (19) | $2NO + O_2 \rightarrow 2NO_2$ | $2.0 \times 10^{-38}$ $[cm^6 \cdot molecule^{-2} \cdot s^{-1}]$ |
| (20) | $NO + O_3 \rightarrow NO_2 + O_2$ | $1.8 \times 10^{-14}$ |
| (21) | $NO + NO_3 \rightarrow 2NO_2$ | $2.6 \times 10^{-11}$ |
| (22) | $NO_2 + O_3 \rightarrow NO_3 + O_2$ | $3.5 \times 10^{-17}$ |
| (23) | $NO_2 + \cdot OH + M \rightarrow HNO_3 + M$ | $4.1 \times 10^{-11}$ |
| (24) | $NO_2 + NO_3 + M \rightarrow N_2O_5 + M$ | $1.9 \times 10^{-12}$ |
| (25) | $N_2O_5 + H_2O \rightarrow 2HNO_3$ | $2.0 \times 10^{-21}$ |
| (26) | $HNO_3 + NH_3 \rightarrow NH_4NO_{3(s)}$ | $10^{-11} \sim 10^{-17}$ (variable) |

$$O_2 + hv \rightarrow \cdot O + \cdot O \tag{27}$$

$$NO_2 + hv \rightarrow NO + O(^3P) \tag{28}$$

To describe the dominant reaction in the NH₄NO₃ formation process in the simulation chamber, O₂ is decomposed into oxygen atoms by 185 nm of UV rays, as shown in Eq. (27), and oxygen atoms and O₂ molecules are combined O₃ as shown in Eq. (4). In addition, through the photolysis reactions of NO₂, it is divided into NO and $O(^3P)$, and $O(^3P)$ is recombined with O₂ to
form O₃ as described above. The third body, M, is any inert gas molecule that can absorb the excess vibration energy and stabilize the formed O₃ so that it does not decompose in the atmosphere, generally N₂ or O₂. A powerful atmospheric oxidant, O₃, reacts with NO to produce NO₂ and O₂. In this way, Eq. (4), (20), and (28) reactions are called photo-stationary states (PSS), and NO, NO₂, and O₃ are maintained in equilibrium (Rothman et al., 2013). HNO₃, one of the essential precursors for forming NH₄NO₃, is produced by the chemical reactions of NO₂ and OH radicals formed by photolysis reactions, as described
in Eq. (23). Although various photochemical reactions that form HNO₃, such as the NO₂ hydrolysis reaction $2NO_2 + H_2O \rightarrow \cdot HONO + HNO_3$, the Eq. (23) reaction is typically dominant reaction during daytime. Gaseous NH₃ and HNO₃ are condensed on the surface of existing particles to form particulate NH₄NO₃.

### 4.3 Effect of UV light intensity on NH₄NO₃ formation

Since sunlight's light intensity varies depending on the weather or season and the reaction rate of precursors varies accordingly,
research on the concentration variation of NO and NH₃ depending on light intensity should be conducted. Therefore, the number of UV-C lamps was changed to control light intensity when NO, NH₃, purified air, and N₂ have the constant mass flow rate, as shown in case 1 of Table 1.





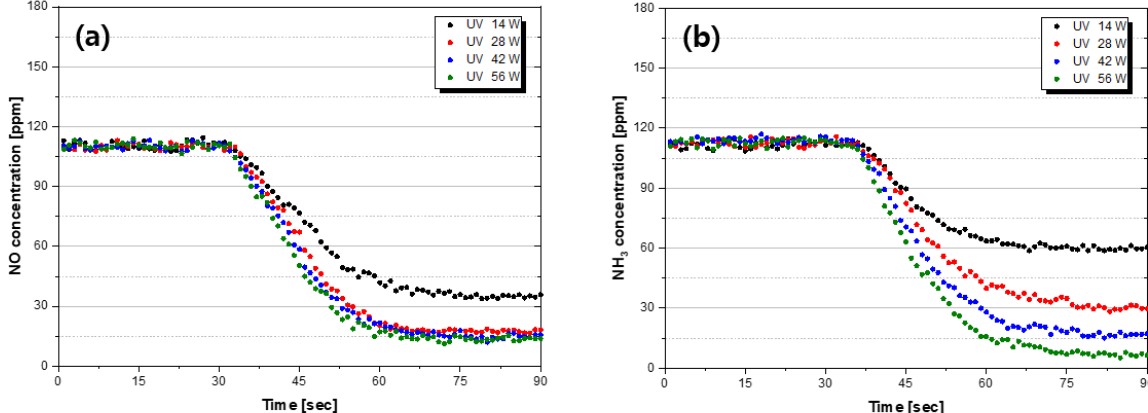

**Figure 3: Time evolution of NO and NH$_3$ concentrations in NO, NH$_3$, H$_2$O, O$_2$, and N$_2$ mixtures at various UV-C light intensities. (black dot: 14W emitted from 1 UV-C lamp, red dot: 28W emitted from 2 UV-C lamps, blue dot: 42W emitted from 3 UV-C lamps, green dot: 56W emitted from 4 UV-C lamps)**

Fig. 3(a) and (b) show NO and NH$_3$ concentration graphs according to the number of applied UV-C lamps. The photochemical reaction test, according to light intensity, was measured while continuously injecting gas for 90 seconds, and the UV-C lamps were operated for 30 to 90 seconds. In the case of NO, under the 14 W condition where one lamp was applied, it decreased by 69% to reach the steady-state for about 42 seconds. Furthermore, under the conditions where UV-C lamps were more applied, it reached a steady-state after decreasing by 85% over approximately 35 seconds. It can be seen that O$_3$ and OH radicals were formed from the decomposition and binding of H$_2$O and O$_2$ by UV light emitted from UV-C lamps, and under conditions with stronger light intensity, the reaction quantity and rate increased due to the enhanced photolysis rate. Gaseous NH$_3$ decreased by about 47, 73, 85, and 94% as UV-C lamps increased. In the case of NO, the maximum NO reduction quantity is almost similar when the light intensity is 28 W or higher, whereas the reduced quantity of NH$_3$ tends to increase gradually. This is attributed to the increase in light intensity, as indicated by Eq. (10), which leads to an increase in ·O, causing the decomposition of NO$_2$ into NO and O$_2$. Additionally, the increased decomposition of H$_2$O into OH radicals by higher light intensity is considered responsible for the conversion of NO$_2$ into HNO$_3$ as described in Eq. (23).

From a temporal perspective, it was observed that there is a slight delay in the decrease of NO and NH$_3$ concentrations after UV light irradiation begins. This delay can be attributed to the time required for the reacted gases to travel from the chamber to the multi-pass cell. Here, the reaction start time represents when the initial concentration decreases by over 3%. The reaction of NO was faster than that of NH$_3$ because NO is reacted faster by O$_3$ and OH radicals, i.e., while NH$_3$ begins to react after HNO$_3$ is formed. In addition, its reaction rate is also considered to be slower than the rate at which NO is converted to NO$_2$ or $\cdot HONO$.



### 4.4 Effect of initial O₄ concentration on NH₄NO₃ formation

In the atmosphere, $O_3$ is generated through the photolysis process of $NO_2$, and $NO_2$ is converted again by reaction with NO. Also, highly reactive hydroperoxyl radical ($\cdot HO_2$) and peroxyl radical ($\cdot RO_2$) produced by volatile organic compounds (VOCs) oxidation convert NO to $NO_2$ (Carslaw et al., 1999). In this sequence of reactions, $RO_2$ and $HO_2$ in the atmosphere determine the net production of ozone. Therefore, since the $O_3$ concentration increases in an environment where VOCs are emitted, it is necessary to study the process of $NH_4NO_3$ photochemical reactions according to various $O_3$ concentrations.

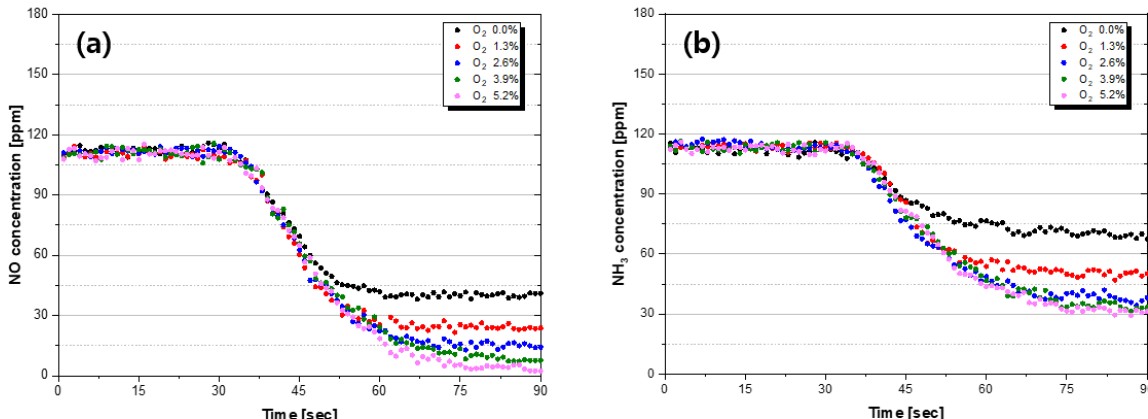

**Figure 4: Time evolution of NO and NH₃ concentrations in NO, NH₃, H₂O, O₂, and N₂ mixtures at various initial O₂ concentrations. (black dot: X_{O2} 0%, red dot: X_{O2} 1.3%, blue dot: X_{O2} 2.6%, green dot: X_{O2} 3.9%, pink dot: X_{O2} 5.2%)**

In this case 2, to confirm the effect of the $O_3$ concentration on photochemical reaction, an experiment was conducted by adjusting the concentration of $O_2$. In other words, since $O_3$ is produced by decomposition and a combination reaction of $O_2$ by UV light emitted from UV-C lamps, the concentration of $O_3$ can be adjusted according to the injected $O_2$ concentration. Fig. 4(a) and (b) are experimental results of NO and $NH_3$ measured according to the concentration of injected $O_2$, respectively. As the $O_2$ concentration is increased from 0 to 5.2%, the conversion rates of NO and $NH_3$ tend to gradually increase to 64, 79, 87, 93, 95%, and 38, 56, 66, 70, and 72%, respectively. In the case of $NH_3$, as conversion rates become similar when the $O_2$ concentration is higher than 2.6%, there is no significant concentration variation above a specific concentration of $O_3$. The reason for this result can be inferred as follows: as the $O_2$ concentration increases, the concentration of $O_3$ also increases, leading to an increasing conversion rate of NO through reactions such as Eq. (19) and (20). However, since the $H_2O$ concentration remains constant, the concentration of OH radicals is limited, preventing sufficient occurrence of the reaction described in Eq. (23). Therefore, the availability of $HNO_3$ required for the reaction in Eq. (26) becomes limited, resulting in $NH_3$ conversion rate and reduction showing a converging pattern over time.





As the $O_2$ concentration increased, the reach time to the steady-state of NO and $NH_3$ gradually increased. It is estimated that this is due to the increase in the reduced quantity of NO and $NH_3$ as the concentration of $O_3$ increases. Even under the condition in which the concentration of $O_2$ expressed by a black dot is 0%, that is, under the condition in which $O_3$ is rare, NO and $NH_3$ were converted by 64% and 38%, respectively. This conversion is caused by OH radicals decomposed from $H_2O$ by UV rays. NO, and OH radicals form $\cdot HONO$, as in Eq. (12), (13) reactions, rather than the dominant reaction during the daytime Eq.

(20). Through this process, it is demonstrated that NOx can form $HNO_3$ even in an environment where $O_3$ is rare. As these reactions are limited compared to the reaction occurring in an environment where $O_3$ is sufficient, it is considered that the conversion rate and reduced quantity of NO and $NH_3$ are relatively small.

## 4.5 Effect of relative humidity on $NH_4NO_3$ formation

In the atmospheric environment, moisture is one of the essential components for the formation of $NH_4NO_3$. $H_2O$ is decomposed

into OH radical through a photolysis reaction or forms $HNO_3$ through a hydrolysis reaction with $N_2O_5$ at night. In addition, the growth of hygroscopic particles such as sulfate and nitrate enhances the diameter of the particles and further degrades the visibility rate when the relative humidity increases (Cao et al., 2012). Therefore, a study is required to understand the effect of relative humidity on $NH_4NO_3$ photochemical reactions in the atmosphere. In case 3, to control the relative humidity, the flow rate of $N_2$ supplied to the $H_2O$ bubble generator heated to 350 K was controlled, and accordingly, an experiment was performed

to measure the concentration variation of NO and $NH_3$. Relative humidity and temperature for each condition were measured by a capacitive probability sensor and PT100 type (TESTO 605i).

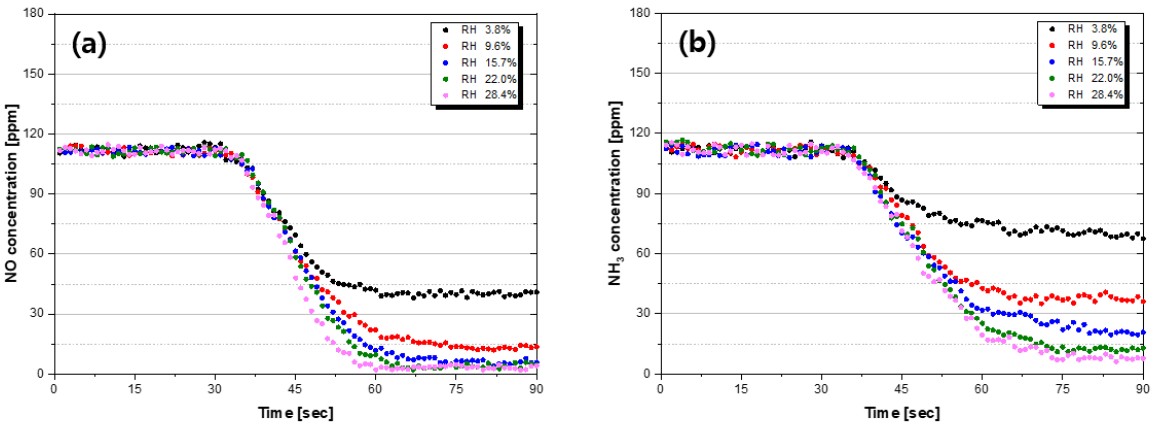

**Figure 5: Time evolution of NO and $NH_3$ concentrations in NO, $NH_3$, $H_2O$, $O_2$, and $N_2$ mixtures at various relative humidity. (black**
**dot: RH 3.8%, red dot: RH 9.6%, blue dot: RH 15.7%, green dot: RH 22.0%, pink dot: RH 28.4%)**

Fig. 5(a) and (b) present the experimental results measuring the concentration fluctuations of NO and $NH_3$ according to the relative humidity. When 300 ppm of NO/$N_2$, $NH_3$/$N_2$ reference gas, purified air, and $N_2$ passing through the $H_2O$ bubble





generator were injected according to each condition in the simulation chamber, relative humidity was measured at 3.8, 9.6,
15.7, 22.0, and 28.4% in sequence. In the case of NO, the conversion proceeded to 64, 88, 95, 96, and 97% as the relative
humidity increased, and $NH_3$ also showed a greater conversion rate than other conditions, such as UV light intensity or $O_3$
concentration. As the relative humidity increases, the amount of OH radical converted from $H_2O$ also increases, and the
reactions of Eq. (12) and (13) occur actively along with the reaction Eq. (20), producing a large quantity of $NO_2$. Additionally,
it is considered that various reactions, such as Eq. (6), (15), (16), and (18), interact in combination. Thus, $NO_2$ forms plenty of
$HNO_3$ because of Eq. (23) sufficient OH radical concentration, and $HNO_3$ leads to $NH_4NO_3$ by reacting with $NH_3$ like Eq. (26).

### 4.6 Effect of initial NO and $NH_3$ concentration on $NH_4NO_3$ formation

NO in the atmosphere is generated during the combustion process from various plants, transportation, and energy systems, and
agricultural activities, such as fertilizer and livestock excrement, primarily produce $NH_3$. This means that a significant amount
of NOx are emitted around factories and downtown areas, whereas $NH_3$ concentrations are high in agricultural and livestock
industries. In such environments, the atmospheric chemical mechanisms operate differently, making studying the effect of
precursor concentrations necessary.

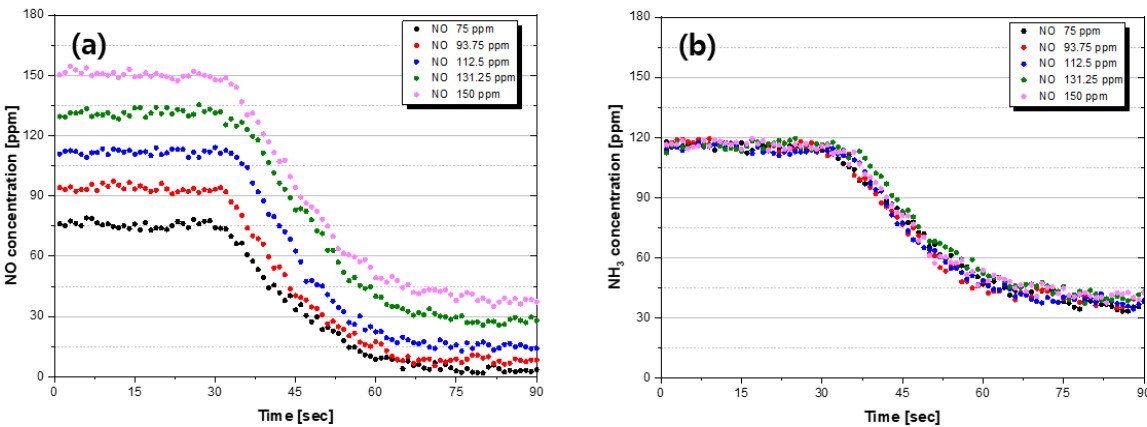

**Figure 6: Time evolution of NO and $NH_3$ concentrations in NO, $NH_3$, $H_2O$, $O_2$, and $N_2$ mixtures at various initial NO concentrations. (black dot: $X_{NO}$ 75 ppm, red dot: $X_{NO}$ 93.75 ppm, blue dot: $X_{NO}$ 112.5 ppm, green dot: $X_{NO}$ 131.25 ppm, pink dot: $X_{NO}$ 150 ppm; when $X_{NH3}$ is fixed at 112.5 ppm)**

In case 4, the concentration fluctuations of NO and $NH_3$ were measured by DAS while controlling the flow rates of NO/$N_2$
and $NH_3$/$N_2$ gases to check how the concentration ratio of NO and $NH_3$ in the atmosphere affects the formation of $NH_4NO_3$.
Fig. 6(a) and (b) show the changes in NO and $NH_3$ concentrations when added while increasing the concentration of NO from
75 to 150 ppm after $NH_3$ is fixed at 112.5 ppm. At this time, the NO conversion rate tends to decrease gradually to
approximately 95, 91, 87, 79, and 75%, while the reduction quantity steadily increases. On the other hand, in the case of $NH_3$,





only a constant quantity is converted to about 65%, regardless of the change in NO concentration. Increasing NO concentration, which means a higher amount of NO participation in Eq. (9), (19), (20), leads to an increased conversion to $NO_2$. However, in
the case of $NH_3$, it is estimated that the constant concentration of $NH_3$ is the result of an unaltered $HNO_3$ amount by the steady concentration of OH radical.

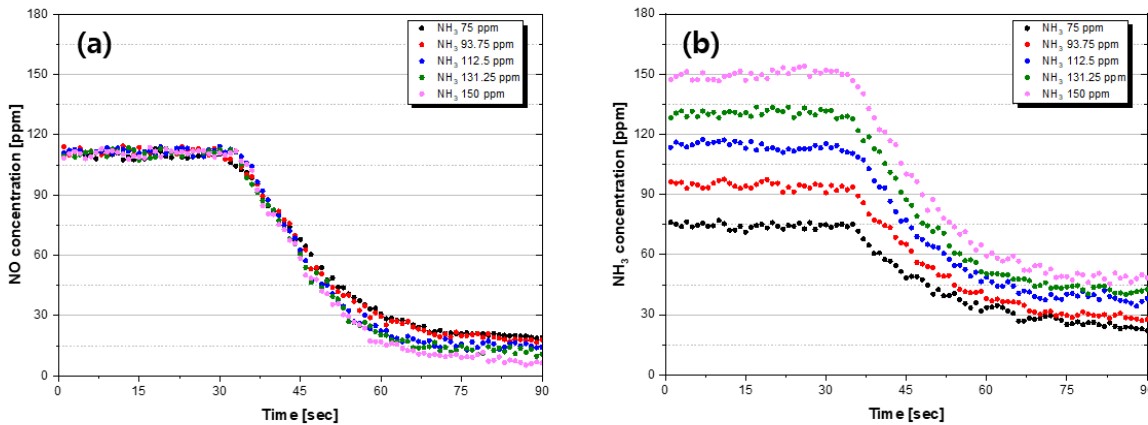

**Figure 7: Time evolution of NO and $NH_3$ concentrations in NO, $NH_3$, $H_2O$, $O_2$, and $N_2$ mixtures at various initial $NH_3$ concentrations.**
**(black dot: $X_{NH3}$ 75 ppm, red dot: $X_{NH3}$ 93.75 ppm, blue dot: $X_{NH3}$ 112.5 ppm, green dot: $X_{NH3}$ 131.25 ppm, pink dot: $X_{NH3}$ 150 ppm; when $X_{NO}$ is fixed at 112.5 ppm)**

On the contrary, Fig. 7 shows the fluctuations in the concentration of NO and $NH_3$ when NO is fixed at 112.5 ppm, and the concentration of $NH_3$ is controlled from 75 to 150 ppm. When the ratio of NO and $NH_3$ concentrations was adjusted to 1:0.67,
0.83, 1, 1.17, and 1.33, the conversion rate of NO was 82 to 93%, which was a relatively narrow variation, and $NH_3$ was also 66 to 69%, almost constant. However, the reduction quantity of $NH_3$ rapidly increases, which is considered to be because as the concentration of $NH_3$ increases, the quantity of its decomposition into $NH_2$ by OH radicals increases, as shown in Eq. (14). The time takes from the reaction start time of NO and $NH_3$ to reach steady-state was shortened as the concentration of $NH_3$ increased. Through this, it was confirmed that the concentration of $NH_3$ partially affected the conversion rate and reaction rate.

**5 Conclusions**

In this study, a small-scale indoor atmospheric simulation chamber was designed to understand the physicochemical processes of $NH_4NO_3$ formation in detail. And UV-C light intensity, $O_3$ concentration, relative humidity, NO, and $NH_3$ initial concentrations were set by experimental conditions, and variation of NO, and $NH_3$ concentrations was measured and analyzed in real-time using laser absorption spectroscopy.





First, $O_3$ and OH radicals were produced through the decomposition and binding of $H_2O$ and $O_2$ by UV rays, and $HNO_3$ was formed through a reaction with NO. In addition, it was confirmed that under conditions with intense UV light, the photolysis ratio increased, increasing both the reaction quantity and reaction rate. The effect of the concentration of $O_3$ on the photochemical reaction is that the concentration of $O_3$ produced by the decomposition and binding reaction increases as the flow rate of $O_2$ injected increases; thereby, the $NO_2$ production ratio increases as in Eq. (20) reaction. The reaction of NO and

$NH_3$ was measured even under rare $O_3$ conditions, as in Eq. (12) and (13) reactions; OH radical generated from $H_2O$ was combined with NO to form $\cdot HONO$ and then $\cdot HONO$ reacted with OH radical again to convert it to $NO_2$. Compared to other conditions, the concentration fluctuations of NO and $NH_3$ change most dramatically according to relative humidity because the mole fraction of OH radicals, which are essential precursors for $NH_4NO_3$ formation, has increased significantly. An experiment was also performed to determine the effect on photochemical reactions according to the mole fraction ratio of NO

and $NH_3$ in the simulation chamber. Regardless of the initial concentration of NO injected, the conversion rate of $NH_3$ remained constant. However, as the initial concentration of $NH_3$ increased, both NO and $NH_3$ reached a steady-state more quickly. Through this, it was confirmed that the concentration of $NH_3$ partially affected the conversion rate and reaction rate. Summarizing these results above, it was confirmed that the higher the UV-C light intensity, $O_3$ concentration, and relative humidity, the better $NH_4NO_3$ was produced through the reduced quantity of NO and $NH_3$. In particular, it was experimentally

demonstrated that the secondary aerosol nucleation and growth process greatly influenced relative humidity.

Unlike previous studies that relied on photochemical reaction results in simulation chambers using sampling methods such as chemiluminescence, NDIR, and gas chromatography, this study demonstrated the potential for transient analysis of the $NH_4NO_3$ formation process using laser absorption spectroscopy. This approach allows for determining of the priority of photochemical reactions and estimating of reaction acceleration or inhibition characteristics of specific factors under various

conditions. It is believed that this methodology can be applied to investigating complex generation and growth mechanisms that have not yet been identified, opening up new avenues for future research in the field.

**Author contributions**

Conceptualization: NJ, CL; Data processing: SL, DK; Formal analysis: DK, MY; Methodology: NJ, SL; Supervision: SS, CL; Writing (original draft preparation): NJ, MY; Writing (review and editing): SS, CL

**Competing interests**

The contact author has declared that none of the authors has any competing interests.



**Acknowledgements**

The authors acknowledge funding from the Industrial Strategic Technology Development Program - Development of technology of Next-generation Intelligence Semiconductor (20023296, Development of Real-Time Continuous Measurement

Equipment for Semiconductor Process Gas Monitoring) funded By the Ministry of Trade, Industry & Energy (MOTIE, Korea).

**Financial support**

This work was supported by the Industrial Strategic Technology Development Program - Development of technology of Next-generation Intelligence Semiconductor (20023296, Development of Real-Time Continuous Measurement Equipment for Semiconductor Process Gas Monitoring) funded By the Ministry of Trade, Industry & Energy (MOTIE, Korea).

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
