# Peer review of "Measurement of NO and NH3 Concentrations in Atmospheric Simulation Chamber Using Direct Absorption Spectroscopy"

_EGUsphere, 2024_

## Author Comment (AC2)

Dear Editor and Reviewer,

We thank the reviewer for their detailed and constructive feedback, which has helped us to identify areas where the manuscript can be significantly improved. Below, we address each comment in detail. Changes made in the manuscript are highlighted and referenced in our responses.

Comment 1.

The title does not capture the essence of the work described. The experiments reported are in a chemical regime that is not appropriate for realistic atmospheric conditions and are based on a chemical reaction volume of only 3.85 L. The term "Atmospheric Simulation Chamber" in the title and manuscript is thus inadequate owing to the high mixing ratios of the reactants and the fact that UV-C was used to drive the photochemistry. The aspect of particle formation through photochemical reactions of NO and $NH_3$ are also not captured by the title.

Response

We acknowledge the reviewer's concern and have revised the title to more accurately reflect the scope of our work. The new title is:

"Real-Time Measurement of NO and $NH_3$ Concentration Variations Using Direct Absorption Spectroscopy in an Indoor Small-Scale Smog Chamber to Analyze $NH_4NO_3$ Photochemical Formation Characteristics"

Comment 2.

The introduction is long-winded and contains many generic statements that are not really specific for the work described later. Statements meant to motivate the topic are dragged out

in some parts, and it is not quite clear where the authors intend to go; the work objectives are not succinctly stated and become apparent only at the very end of the introduction.

Response

We appreciate this comment. We have shortened the introduction by removing generic statements and focusing on the specific context of our study.

Comment 3.

The technological advances of the work are limited. Direct absorption spectroscopy in all its experimental realizations (differential optical absorption spectroscopy, multi-pass cells, cavity-enhanced variants, TDLAS) has been used for decades in atmospheric sciences for trace gas detection. While the setup shown in Figure 1 is custom-designed for the current study and appears to bear some novel aspects, TDLAS itself is a well-known and widely used approach for the detection of trace gases. Consequently Chapter 2 on the "theoretical background" does not contain new information; it could be significantly shortened.

Response

We appreciate the reviewer's feedback regarding the content of Chapter 2. In response, we have significantly shortened this section by focusing only on the theoretical aspects directly relevant to our study. Redundant descriptions of well-established methods, such as the general principles of direct absorption spectroscopy, have been removed.

Comment 4.

* It would be good to give the full dimensions of the reaction chamber. A volume of 3.85 L with a surface-area-to-volume ratio of 1.75 $m^{-1}$ implies a surface area of only 67 $cm^2$. Which

appears very small, especially since the chamber contains 4 UV lamps, whose surface area should also be taken into account in the ratio. What about area and volume of the multi-pass cell?

* The nature of the deflection mirrors in the White cell after the entrance and before the exit aperture is not clear. How are two incident parallel light rays reflected in two different directions. What feature do these "steering" mirrors have – or are there two mirrors used?

* Section 3 is missing a number of important experimental parameters that describe the experimental conditions, such as duty cycle, integration time, residence time of gas mixtures, purity of chemicals, cleaning and calibration procedures, inlet losses to the multi-pass cell, to name a few.

* UV-C does not cover the vacuum UV < 180 nm ("···100 to 280 nm···"). Acronym HHL (L.124) and PSS (L. 213) not used elsewhere in the text.

Response

We thank the reviewer for these detailed comments and have made the following revisions to the manuscript:

* Dimensions and surface-area-to-volume ratio:

Chamber surface area and volume: 0.228 m$^2$ & 3.85 L

Multi-pass cell surface area and volume: 0.211 m$^2$ & 5.65 L

1ea UV lamp surface area and volume: 0.0168 m$^2$ & 0.062 L

These details have been added to the manuscript to clarify the experimental setup.

* Deflection mirrors in the White cell:

Although the diagram in the manuscript illustrates the lasers as parallel for simplicity, in the actual experimental setup, the incident angles of the two laser beams are different. This clarification has been added to the text to provide a more accurate description of the optical

configuration.

* Experimental conditions:

In the revised manuscript, the experimental parameters mentioned by the reviewer have been added and are described in detail.

* Use of UV-C light:

UV-C was chosen due to its short wavelength and high-energy photon output, which enables selective induction of specific reactions such as the photolysis of $NO_2$ and the decomposition of $O_3$. This narrow wavelength range facilitates precise analysis of the contributions of specific reaction pathways. Given the study's focus on ozone decomposition, NOx and $NH_3$ interactions, and photochemical radical production, the use of UV-C light is highly appropriate. The revised manuscript includes a detailed explanation of the rationale for using UV-C wavelengths.

We believe these revisions address the reviewer's concerns comprehensively and improve the clarity and scientific rigor of the manuscript.

Comment 5.

The sub-section (4.1) on selecting an appropriate absorption line for detection seems to exclusively contain HITRAN and simulated data. The spectra shown in Figure 2 were seemingly not measured by the authors – if they were, this needs to be made clear. The material in this section (4.1) can be shortened significantly and/or can be largely placed in the supplementary material or into an appendix. Spectra (or even data) that were measured to retrieve mixing ratios stated later in the result section are however not shown – they should be included in the manuscript also in the context of a meaningful error discussion.

Response

We appreciate the reviewer's comments regarding sub-section (4.1). The content in this

section is indeed based on HITRAN and simulation data, as it describes the process of selecting absorption lines that minimize interferences from precursor or intermediate species that may arise during the photochemical reactions of NO and $NH_3$. Given the importance of selecting wavelengths free from such interferences for accurate measurements, we provided detailed explanations in the original manuscript.

In response to the reviewer's suggestions, we have made the following changes:

The detailed HITRAN data has been moved to the supplementary material, and the section on absorption line selection has been summarized for conciseness.

Mixing ratios were calculated using the flow rates controlled by the mass flow controllers (MFCs). To address measurement errors and uncertainties, actual absorption data have been added to the manuscript for clarity and improved discussion.

We believe these revisions effectively address the reviewer's concerns and enhance the focus and scientific rigor of sub-section (4.1). Thank you for highlighting this point.

Comment 6.

Case 5 in Table 1 is not explicitly mentioned in the text in section 4.

Response

Thank you for pointing out the lack of explicit mention of Case 5 in Section 4. The discussion related to Case 5 is indeed presented in the context of Figure 7, where the effects of varying $NH_3$ concentrations on $NH_4NO_3$ formation are analyzed. However, we realize that this connection was not clearly stated in the text.

To address this, we have revised the manuscript to explicitly reference Case 5 in the discussion of Figure 7. Specifically, we have added the following clarification:

*"In Case 5, the initial $NH_3$ concentration was varied to analyze its impact on $NH_4NO_3$ formation.*

*Figure 7 presents the time evolution of NO and NH₃ concentrations under these conditions."*

This revision ensures that readers can directly associate Case 5 with the corresponding experimental setup and results.

Comment 7.

In Table 2 the column for "references" is empty – i.e. no references are stated. Reactions are numbered in column 1 according to the equation numbering. This is not uniform (see eq. (27) and (28)).

Response

Thank you for your observation. Table 2 has been moved to the supplementary material to streamline the main text. Additionally, the reactions previously listed in Table 2 are now presented directly as equations in the main text for better clarity and uniformity.

Comment 8.

The reactions in Table 2 are used in explanations and interpretations in sections 4.3 – 4.6, however, the modelling of data (e.g. MCM or Facsimile) was not attempted. All attempts to explain the results are merely qualitative and not quantitative. Other intermediates or resulting aerosol were not detected in order to develop a quantitative model describing the results outlined.

Response

Thank you for your valuable feedback regarding the interpretation and modeling of the results. We acknowledge that the current study primarily provides qualitative explanations for the observed results without employing quantitative modeling tools such as MCM or Facsimile.

This approach was chosen due to the study's primary focus on real-time measurement of NO and $NH_3$ concentrations and the experimental investigation of variables influencing $NH_4NO_3$ formation.

Regarding the detection of other intermediates or resulting aerosols, we recognize that such data would be critical for developing a comprehensive quantitative model. However, this aspect was beyond the scope of the current study, which was designed to focus on the gas-phase precursors and their conversion under controlled conditions.

We agree that integrating quantitative modeling and detecting intermediates would greatly enhance the understanding of the processes described. These aspects are part of our future research plans, where we aim to extend the experimental framework and incorporate advanced detection techniques and modeling tools to provide a more detailed and quantitative interpretation.

Comment 9.

The conclusion section is somewhat repetitive and merely mostly outlines the performed experiments and results again and not what can be learnt from the new data.

Response

Thank you for pointing out the limitations in the conclusion section. We agree that the section can be improved to focus more on the broader implications and key learnings from the new data, rather than reiterating the experimental details and results.

To address this, we have revised the conclusion section to emphasize the insights gained from the study, such as the critical role of UV-C light intensity, relative humidity, and precursor concentrations in $NH_4NO_3$ formation, and the utility of real-time laser absorption spectroscopy in understanding these processes. Additionally, we have highlighted how these findings

contribute to the broader understanding of secondary aerosol formation mechanisms and their potential applications in atmospheric chemistry research and pollution control strategies. We hope this revised conclusion provides a clearer synthesis of the study's contributions and relevance.

Comment 10.
Finally the manuscript would certainly also benefit from some revision concerning the use of the English language.

Response
Thank you for your feedback regarding the use of English language in the manuscript. We have carefully reviewed and revised the manuscript to improve the clarity, grammar, and overall readability. We believe these efforts have significantly enhanced the manuscript and addressed the issues raised.
We appreciate your suggestion and are confident that the revised version meets the expected standards.